# Sand Burial, Rather than Salinity or Drought, Is the Main Stress That Limits the Germination Ability of *Sophora alopecuroides* L. Seed in the Desert Steppe of Yanchi, Ningxia, China

**DOI:** 10.3390/plants12152766

**Published:** 2023-07-25

**Authors:** Jingdong Zhao, Chaoyi Shi, Danyu Wang, Yuanjun Zhu, Jiankang Liu, Hanzhi Li, Xiaohui Yang

**Affiliations:** 1Breeding Base for State Key Lab. of Land Degradation and Ecological Restoration in Northwestern China/Key Lab. of Restoration and Reconstruction of Degraded Ecosystems in Northwestern China of Ministry of Education, Ningxia University, Yinchuan 750021, China; 2Institute of Desertification Studies, Chinese Academy of Forestry, Beijing 100091, China; 3Institute of Ecological Conservation and Restoration, Chinese Academy of Forestry, Beijing 100091, China

**Keywords:** sand burial, salinity, drought, seed germination, *Sophora alopecuroides* L.

## Abstract

Global change and environmental pollution have reawakened ecologists to the great threat of multi-stress interactions to different growth stages of plants. *Sophora alopecuroides* L., a dune plant, has been widely studied for its medicinal components and strong salinity tolerance. *S. alopecuroides* seeds, obtained from the desert steppe of Yanchi, Ningxia, China, were used to analyze the effects of sand burial, salinity, drought, and their interactions on seed germination (germination percentage, germination energy, and germination index). The results showed that sand burial and salinity stress had significant effects on the seed germination ability of *S. alopecuroides*, and drought stress had no significant effect, but the interaction of the three stresses had a significant effect. Under different drought-stress treatments, the interaction of no sand burial and a certain degree of salinity stress significantly improved the germination ability of *S. alopecuroides*, and the overall intensity of the effects of the three stresses showed that sand burial > salinity > drought. Considering the germination percentage, germination energy, and germination index of *S. alopecuroides* under various stress interactions, the treatment of no sand burial × 1% soil saline-alkali content × 18–20% soil water content was adopted to maximize the germination ability of *S. alopecuroides*. In the desert steppe area of Yanchi, Ningxia, sand burial stress was still the most limiting factor for seed germination of *S. alopecuroides*, and soil saline-alkali content should be increased moderately, and soil moisture should be ensured to obtain the best germination ability.

## 1. Introduction

Biotic and abiotic stresses are the two primary factors that limit plant growth. Biotic stresses mainly involve herbivores and human activities such as feeding, trampling, mowing, and plowing, while abiotic stresses include wind-blown sand, salinity, drought, frosts, fire, etc. [1,2,3]. In desert steppe areas, the harsh wind-blown sand often results in a sand layer covering the soil surface to a certain depth. Overcoming this sand burial stress becomes essential for seed germination and seedling growth for many sandy species [4]. Moreover, desert steppes are characterized by scarce precipitation and high soil evaporation [5]. The severity of soil salinity and the frequency of drought are also significant challenges [2,6]. In the complex and harsh environments, deep sand burial, high salinity, elevated pH, and low water availability often coexist and exert synergistic effects on plants, particularly on seed germination. These combined stressors may prove to be more detrimental to plant growth than individual stresses alone [3,7].

Deeper sand burial may force seed dormancy, leading to the formation of a soil seed bank that may germinate and grow when the sand layer thins out [4]. Alternatively, newly germinating seedlings within the soil that are unable to reach the soil surface will perish [8]. On the other hand, sand burial can play a crucial role in seed germination by reducing the contact area between seeds and air, maintaining high humidity, protecting seeds and seedlings from extreme air temperatures in harsh climates, and isolating them from predators [8]. The magnitude and direction of the effect of sand burial often vary depending on the depth of burial and the species involved [4].

Salinity is an important abiotic stressor affecting agricultural production and the ecological environment [7,9,10], which may affect all the major processes, such as seed germination, growth, photosynthesis, water relation, nutrient imbalance, oxidative stress, and yield. Especially during the seedling and early vegetative growth stage, the growth is very sensitive to soil salinity [9,11]. In addition, salinization also affects a range of environmental interactions that disrupt water infiltration and soil water-storage capacity, leading to increased water runoff and erosion [11,12]. However, in nature, salt stress and alkaline stress often occur simultaneously; when a salt soil contains HCO_3_^−^ and/or CO_3_^2−^, which elevate soil pH, plants experience damage from both salt and alkaline stresses [13], and their synergistic effects can be more harmful to the plant than stress alone [7]. In previous studies, salt stress has received more attention than alkaline stress or saline-alkali mixed stress [7,14], which limits the understanding of seed germination strategies in saline-alkali habitats.

Drought stress affects seed germination and seedling development by inducing osmotic stress [15]. Physiologically, drought stress can lead to turgor loss, reduced water potential, decreased stomatal conductance, and, therefore, reduced internal CO_2_ concentration and the net carbon fixation rate, which in turn leads to reduced plant growth rate and biomass [16]. However, some desert plants showed unique tolerance and response mechanism to drought stress, such as *Pinus sylvestris* var. *mongolica* from Hulunbuir sandy land and *Eremosparton songoricum* from Gurbantunggut Desert, China, and a certain degree of drought stress was conducive to seed germination, which may be the result of long-term adaptation to the natural environment [17,18].

*Sophora alopecuroides* L., a leguminous herb widely distributed in the desert and semi-desert areas of northern China is often utilized to prevent land desertification and reduce soil erosion due to its excellent tolerance to wind-blown sand, salinity, and drought. The alkaloids extracted from the aerial parts and seeds of *S. alopecuroides*, such as marine and oxymatrine, have been extensively studied and developed into new drugs [19]. Additionally, *S. alopecuroides* serves as a valuable resource for livestock forage, green manure, windbreaks, and nectar, demonstrating its high conservation and exploitation value [20]. Due to the interactive effects of biological and abiotic factors, the desert steppe is increasingly vulnerable, and poor seed germination and seedling establishment are considered important factors limiting the subsequent growth and yield of *S. alopecuroides* [5,21]. However, there is still a significant research gap regarding how this herb responds to multi-stressor combinations during seed germination. Therefore, a greenhouse-based seed germination experiment was conducted using *S. alopecuroides* seeds collected from the desert steppe (Yanchi County, Ningxia) to explore the effects of multifactorial stress combinations on seed germination percentage, germination energy, and germination index. This study aims to (1) investigate the effects of sand burial, salinity, and drought stress on the germination ability of *S. alopecuroides*, (2) compare the effects of these stressors on the seed germination ability of *S. alopecuroides* seeds, and (3) determine the optimal interaction of factors for the germination of *S. alopecuroides* in the local area. Understanding the germination ability of *S. alopecuroides* seeds under multi-stress combinations is crucial not only to the development of vegetation restoration programs in desert steppe areas but also to the conservation and management of the local ecological environment.

## 2. Results

### 2.1. Effects of Sand Burial, Salinity, and Drought Stress on Germination Percentage, Germination Energy, and Germination Index of S. alopecuroides Seeds

The analysis of three-way ANOVA showed that the germination percentage, germination energy, and germination index of *S. alopecuroides* seeds were significantly affected by sand burial (A) and salinity (S) stress (*p* < 0.001), and the drought (D) stress had no significant effect on seed germination. The interaction of A × S stress had significant effects on germination percentage, germination energy, and germination index (*p* < 0.001). The interaction of A × D stress had significant effects on the germination index (*p* < 0.05) but insignificant effects on germination percentage and germination energy. The interaction of S × D stress had significant effects on germination energy (*p* < 0.05) but insignificant effects on germination percentage and germination index. The interaction of A × S × D stress had significant effects on germination percentage (*p* < 0.01), germination energy (*p* < 0.001), and germination index (*p* < 0.05) (Table 1).

With the increase in sand depth, the germination percentage, germination energy, and germination index of *S. alopecuroides* seeds showed a gradually decreasing trend. Under different drought and salinity stress, the germination ability of *S. alopecuroides* seeds under 0 cm and 2 cm sand depth was higher, while the germination ability under 4 cm and 6 cm sand depth was lower. With the increase of salinity stress, the germination percentage, germination energy, and germination index of *S. alopecuroides* seeds were firstly increased and then decreased. The germination ability of seeds under moderate salinity stress was higher, while lower under no salinity stress and high salinity stress. The germination percentage and germination index of *S. alopecuroides* seeds were the highest in A1 × S3 × D1 interaction, but the germination energy was highest in A1 × S5 × D5 treatment (Figure 1).

### 2.2. Comparison of Influence Intensity among Different Stressors

Stepwise regression analysis was used to compare the limiting role of sand burial, salinity, and drought stress on the germination ability of *S. alopecuroides* seeds (Table 2). In the absolute value of the stressors, the standard regression coefficients (*β*_1_) of sand burial were higher than those of salt alkali (*β*_2_), and the standard regression coefficients (*β*_3_) of drought were all 0. Therefore, based on the germination ability of *S. alopecuroides* seeds, the effect of sand burial was greater than that of salinity, and drought had no effect.

### 2.3. Comprehensive Evaluation of Germination Ability Based on Principal Component Analysis

After the principal component analysis, the KMO value is 0.734 > 0.500, and the corresponding score can be calculated by principal component analysis. According to the criteria of initial eigenvalue > 1 and cumulative contribution rate > 90%, one principal component score was extracted, which could respond to 92.698% of the information content. The best germination ability of *S. alopecuroides* seeds was achieved when the stress treatment was no sand burial × 1% soil saline-alkali content × 18–20% soil water content (Table 3).

## 3. Discussion

Germination percentage, germination energy, and germination index are important indicators to characterize seed germination; both former parameters are essential to assess the seed germination ability and quality grade, while the last one expresses seed vigor [5]. Seed germination and seedling growth of sand dune plants need to overcome the harsh wind-blown sand conditions. Therefore, sand burial is an important stress factor that restricts the growth, survival, and distribution of sand dune plants [4,22]. Previous studies have revealed that sand burial can have both positive and negative effects on seed germination and seedling emergence in desert steppe ecosystems [8,23]. In this study, with the increase in sand burial depth, seed germination percentage, germination energy, and germination index exhibited a decreasing trend, and the top 10 combined treatments all excluded sand burial, indicating that sand burial was the most critical factor inhibiting the germination of *S. alopecuroides* seeds. Tao et al. found that the stored nutrients within the seeds were the sole source of energy during seed germination and seedling establishment. Under sand burial, seedlings may consume a large amount of energy inside the seeds before breaking through the sand layer [24]. Therefore, the overall germination ability of *S. alopecuroides* seed in this study was low under sand burial stress and showed a gradual decline with the increase of sand burial depth. It is worth noting that, unlike field experiments, indoor controlled experiments avoid the interference of many biotic and abiotic stresses that can interact with sand burial, such as interference from seed predators and wind action [25,26], which are important factors affecting the assessment of sand burial stress. Therefore, large-scale and long-term field experiments should be actively pursued in future seed germination studies to explore the germination mechanism of *S. alopecuroides* in the actual field environment.

In this study, six salinity gradients were set by simulating natural conditions in Yanchi, Ningxia. The results showed that moderate salinity stress may be beneficial to seed germination. It had been reported that a certain concentration of coupled salt-drought could improve the germination percentage, germination energy, and germination index for four common desert plants in the Gurbantunggut Desert, northern China, and improve their tolerance to the arid saline-alkali environment [5]. It has been shown that the seeds of *S. alopecuroides* in the dryland of western Iran, characterized by moderate salt tolerance during germination, could germinate well in a wide range of pH and had a high tolerance to various abiotic stresses, which also led to the rampant in the local agricultural system [27]. In recent years, with the development of genomics, scientists have identified thousands of differentially expressed genes (DEGs) in *S. alopecuroides* in salt, alkali, and drought-stressed environments by transcriptome sequencing to cope with complex and harsh wild environments [28,29]. Therefore, *S. alopecuroides* seeds have certain salinity tolerance, and moderate saline-alkali stress is conducive to seed germination.

Both ANOVA and stepwise regression analysis results showed no significant effect of drought stress on seed germination, which may be due to the strong drought tolerance of *S. alopecuroides* seeds [19,20,27]. These tested seeds were sourced from desert steppe areas, and the lowest water stress gradient set in the experiment may not reach the minimum threshold to limit the germination of *S. alopecuroides* seeds. In addition, another reason may be that the drought gradient range of the experimental setup is too small, and the gradient range of drought stress can be increased in future experiments to obtain more accurate experimental results. However, under the treatment of the highest germination ability of *S. alopecuroides* seeds, the soil moisture content was 18–20%, which indicated that although drought stress did not play a significant role, adequate water conditions could still promote seed germination, especially in the field with the high temperature and wind-blown sand complex habitat, the negative effects of drought may be amplified [30,31]. Xu et al. found that both photosynthetic rate and water-use efficiency decreased with increasing temperature at night, especially under severe water stress conditions, suggesting that nocturnal warming significantly aggravated the adverse effects of soil water stress [32]. Therefore, keeping higher soil moisture may be an effective way to improve the seed germination ability of *S. alopecuroides* in arid and semi-arid desert steppe. Future studies should explore the response of *S. alopecuroides* seeds to drought stress by expanding the drought gradient.

The effects of single-stress and multifactorial stress combinations on plants and soil are quite different [33,34,35], and the responses of plants to multifactorial stress combinations are unique, which cannot be inferred from the responses of plants to single-factor stress [36]. Unfortunately, few studies have considered the effects of abiotic stress with three or more factors on seed germination, and successful germination and initial establishment of seeds in extreme habitats often require overcoming multiple and simultaneous stress factors [37]. In this study, the interaction of sand buried, salinity, and drought had significant effects on seed germination percentage, germination energy, and germination index, but the effect of sand burial stress was significantly stronger than that of salinity stress, while the effect of drought stress was not significant. According to the multifactorial stress mechanism, although each of the different stresses, applied individually to plants, has a negligible effect on their growth and survival, the accumulated impact of multifactorial stress combination on plants is detrimental [34,38]. Our research further indicates that the multifactorial stress mechanism is not only applicable to the plant growth stage but may also be applied to the seed germination stage. Therefore, in complex field environments that include multiple abiotic and biotic stressors with high biodiversity, the relevance and certainty of conclusions drawn from single-factor control experiments are questionable and should be used with caution, and future research should actively pursue long-term multifactorial stress control experiments in the field.

## 4. Materials and Methods

### 4.1. Study Area

This study was performed in a greenhouse at the Institute of Desertification Studies, Chinese Academy of Forestry. Mature seeds of *S. alopecuroides* were collected from plants growing in Yanchi County, Ningxia Autonomous Region, China (37°50′07.516′′ N, 107°25′23.873′′ E) in September 2021 (Figure 2), where mean annual rainfall is about 285.6 mm, mean annual temperature is 8.2 °C, and the area is dry all year with high evaporation and aeolian sand. The soil type is sierozem soil with very low organic matter content and poor nutrients. Plants are mainly xerophytic herbs, semi-shrubs and shrub plants, and the dominant species are *S. alopecuroides*, *Astragalus melilotoides,* and *Stipa caucasica* subspecies *glareosa*, etc. [39].

### 4.2. Experimental Design

To break seed dormancy and maximize seed germination potential [41], this study explored the best pretreatment method for the germination of *S. alopecuroides* seeds before the formal experiment began. Firstly, the seeds of *S. alopecuroides* seeds were selected and set up for the interactive treatment experiment of sulfuric acid (0%, 65%, 75%, 85%, 95%) immersion and water bath (20 °C, 40 °C, 60 °C, 80 °C) immersion. The seeds were firstly soaked with different concentrations of sulfuric acid for 20 min, then washed with water for 10 min to wash off the surface sulfuric acid, followed by constant immersion in distilled water at different temperatures for 20 min. Secondly, the treated seeds were gently wiped dry, and the Petri dish germination method was adopted. Two pieces of filter paper were laid on the bottom of the Petri dish; the seeds were covered with a layer of filter paper, and the filter paper was fully soaked with distilled water. When the germination number did not increase for three consecutive days, the germination test could be considered to be over. The pre-experiment lasted for 14 days. Finally, the pretreatment method of an 85% sulfuric acid × 20 °C water bath was selected by considering seed germination percentage, germination energy, germination index, hypocotyl length, radicle length, seedling length, seedling fresh weight, and water content.

In June 2022, after pretreatment of *S. alopecuroides* seeds, plastic flowerpots with a height of 16.5 cm and a diameter of 13.5 cm were selected as planting containers. The initial medium for all treatments was a mixture of 10 cm nutritive soil and perlite with a volume ratio of 4:1, weighing about 1.5 kg. Then, 30 healthy seeds were placed in each pot and lightly covered with thin soil. Different combinations of sand burial, salinity, and drought stress were set (Figure 3). The sand burial stress included four levels, with sand burial depth of 0 cm, 2 cm, 4 cm, and 6 cm, respectively, recorded as A1, A2, A3, and A4. The specific operation method was to cover the top of the soil mixture with the corresponding thickness of river sand. The salinity stress was divided into six levels, and 0 g, 7.5 g, 15 g, 22.5 g, 30 g, and 37.5 g of mixed salinity were infused into the soil by means of saline-alkali solutions to reach the expected soil saline-alkali content, where the molar ratio of NaCl, Na_2_SO_4_, and NaHCO_3_ was 1:2:1, recorded as S1, S2, S3, S4, S5, and S6. The soil saline-alkali content was 0%, 0.5%, 1.0%, 1.5%, 2.0%, and 2.5%. The drought stress included five levels with a soil water content of 18–20%, 14–16%, 10–12%, 6–8%, and 2–4%, respectively, recorded as D1, D2, D3, D4, and D5. Only fresh water was used for irrigation, and soil moisture levels of all treatments were maintained using the weighing method. A total of 120 multifactor stress treatments were tested, and each treatment had three replicates. 

### 4.3. Data Collection and Analysis

The germination number was counted daily during the germination stage, and the germination trial ended 21 days after sowing. The germination percentage, germination energy, and germination index are calculated as follows [5,42]:

Germination percentage = *n*/*N*·100%, where *n* is the number of normally germinated seeds, and *N* is the total number of tested seeds. Germination energy = *n_p_*/*N*·100%, where *n_p_* is the number of seeds germinated at the peak of germination and N is the total number of tested seeds. Germination index = ∑ *G_t_*/*D_t_*, where *G_t_* is the number of germinated seeds in t time, *D_t_* is the corresponding number of germination days.

Statistical analyses were conducted using SPSS 26.0 software (SPSS Inc.; Armonk, NY, USA). Three-way ANOVA and Duncan multiple contrasts were conducted for significance analysis. For quantifying the effects of sand burial, salinity, and drought on seed germination, stepwise regressions were performed. Principal component analysis (PCA) is a multivariate technique used for data reduction; it is realized through the factor analysis module of SPSS, in which the method of extracting factors selects the principal component, which was used to comprehensively rank the germination ability of *S. alopecuroides* seeds under different combinations of stress factors. Origin 2021 (OriginLab Corporation, Northampton, MA, USA) was used to generate figures.

## 5. Conclusions

The adaptation of plants to multifactorial stress interaction during seed germination has a significant impact on their later growth and yield, which is crucial to maintain economic and ecological security in arid and semi-arid regions. A study on the seed germination characteristics of *S. alopecuroides* seeds under multifactorial stress combinations in desert steppe areas revealed that drought stress had no significant effect on seed germination but a significant effect when treated interactively with sand burial and salinity stresses. Under different drought stress treatments, no sand burial and moderate salinity stress were more favorable for *S. alopecuroides* seed germination, and the best germination ability was achieved when the stress treatment was no sand burial × 1% soil saline-alkali content × 18–20% soil water content. In the desert steppe area of Yanchi, Ningxia, sand burial, rather than salinity or drought, was the main stress limiting the germination ability of *S. alopecuroides* seeds. This study provides important support for seed cultivation, vegetation restoration, and subsequent ecological protection of *S. alopecuroides* in a desert steppe.

## Figures and Tables

**Figure 1 plants-12-02766-f001:**
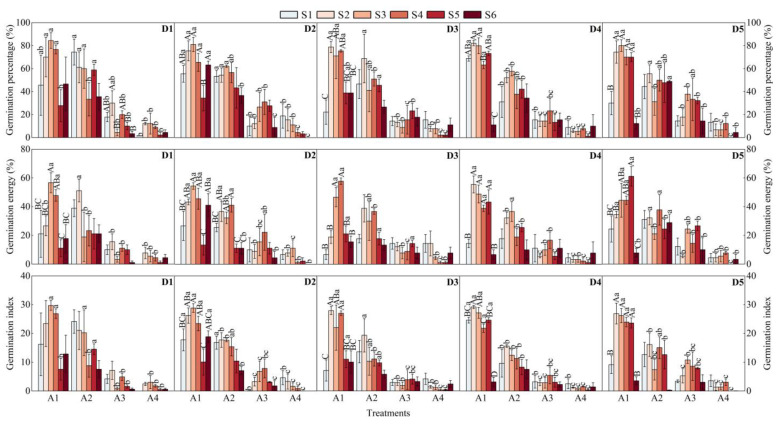
Effects of sand burial, salinity, and drought stress on germination percentage, germination energy, and germination index of *S. alopecuroides* seeds. Different lowercase letters indicate that under the same drought and salinity stress, the germination percentage, germination energy, and germination index under different sand burial stress are significant (analyzed by one-way ANOVA; *p* < 0.05). Different uppercase letters indicate that under the same drought and sand burial stress, the germination percentage, germination energy, and germination index under different salinity stress are significant (analyzed by one-way ANOVA; *p* < 0.05). Sand burial (sand depth): A1 0 cm, A2 2 cm, A3 4 cm, A4 6 cm; salinity (soil saline-alkali content): S1 0%, S2 0.5%, S3 1%, S4 1.5%, S5 2%, S6 2.5%; drought (soil moisture content): D1 18–20%, D2 14–16%, D3 10–12%, D4 6–8%, D5 2–4%.

**Figure 2 plants-12-02766-f002:**
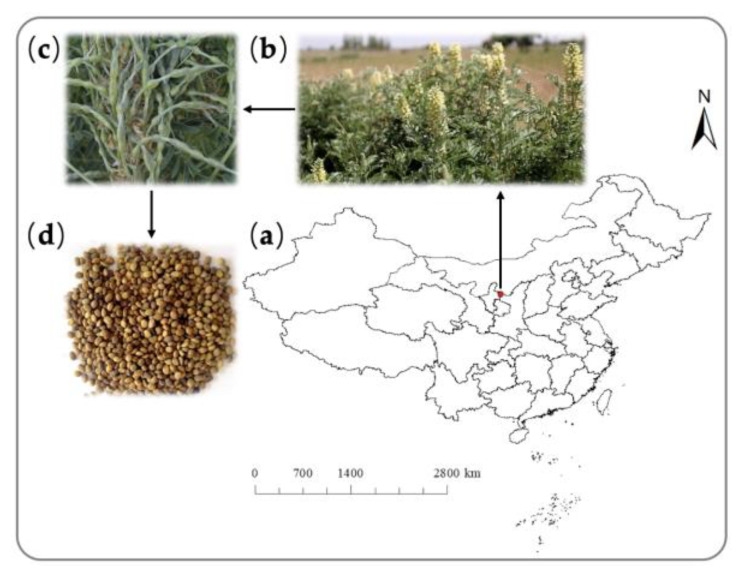
*S. alopecuroides* seeds collection site and plant characteristics [40]: (**a**) seeds collection site, (**b**) flowering branches and habitats, (**c**) fruits, and (**d**) seeds (bean).

**Figure 3 plants-12-02766-f003:**
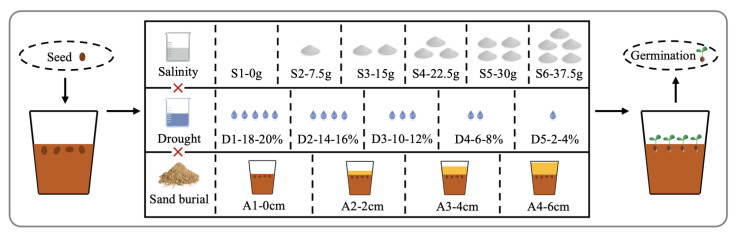
Experimental design diagram. Sand burial (sand depth): A1 0 cm, A2 2 cm, A3 4 cm, A4 6 cm; salinity (soil saline-alkali content): S1 0%, S2 0.5%, S3 1%, S4 1.5%, S5 2%, S6 2.5%; drought (soil moisture content): D1 18–20%, D2 14–16%, D3 10–12%, D4 6–8%, D5 2–4%.

**Table 1 plants-12-02766-t001:** Three-way ANOVA of effects of sand burial, salinity, drought, and their interactions on germination characteristics of *S. alopecuroides* seeds.

Factors	df	Germination Percentage (%)	Germination Energy (%)	Germination Index
Sand burial (A)	3	217.574 ***	149.776 ***	241.899 ***
Salinity (S)	5	11.947 ***	16.147 ***	21.077 ***
Drought (D)	4	0.833 ^NS^	1.716 ^NS^	1.192 ^NS^
A × S	15	4.871 ***	7.111 ***	7.205 ***
A × D	12	1.054 ^NS^	1.239 ^NS^	2 *
S × D	20	0.806 ^NS^	1.843 *	1.44 ^NS^
A × S × D	60	1.608 **	2.209 ***	1.404 *

Note: Numbers are *F*-values significant at * *p* < 0.05, ** *p* < 0.01, *** *p* < 0.001; ^NS^ = non-significant.

**Table 2 plants-12-02766-t002:** Results of stepwise regression between each index and the three stress factors (sand burial, salinity, and drought).

Index	Mode	*R* ^2^	ANOVA Text	*β* _1_	*β* _2_	*β* _3_
Germination percentage (%)	*Y* = 86.589 − 18.459*X*_1_ − 2.089*X*_2_	0.558	*p* < 0.001	−0.736	−0.127	/
Germination energy (%)	*Y* = 48.228 − 10.07*X*_1_ − 1.144*X*_2_	0.421	*p* < 0.001	−0.639	−0.111	/
Germination index	*Y* = 29.023 − 6.312*X*_1_ − 1.077*X*_2_	0.563	*p* < 0.001	−0.726	−0.189	/

Note: *X*_1_ sand burial, *X*_2_ salinity. *β*_1_, *β*_2_ standardized regression coefficients corresponding *X*_1_, *X*_2_. *β*_3_ is the standardized regression coefficient of drought stress. Standardized regression coefficient represents stress level of stress factor. The greater the absolute *β* value, the stronger effect of the stress factor on germination percentage or germination energy, or germination index.

**Table 3 plants-12-02766-t003:** Ranking of the germination ability of *S. alopecuroides* seeds treated with different combinations of stress factors.

Ranking	Sand Burial	Salinity	Drought	Score
	Sand Burial Depth (cm)	Saline-Alkali Content (%)	Water Content (%)	
1	0	1	18–20	2.0884
2	0	0.5	6–8	2.0215
3	0	1	14–16	1.9720
4	0	1.5	10–12	1.9008
5	0	1	6–8	1.7915
6	0	2	2–4	1.7720
7	0	1.5	18–20	1.7188
8	0	1	2–4	1.6725
9	0	0.5	14–16	1.5962
10	0	2	6–8	1.5112

## Data Availability

All data are presented in the main text.

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
