# Peer review of "Sand Burial, Rather than Salinity or Drought, Is the Main Stress That Limits the Germination Ability of Sophora alopecuroides L. Seed in the Desert Steppe of Yanchi, Ningxia, China"

_plants, 2023, doi:10.3390/plants12152766_

Round 1

Reviewer 1 Report

OVERALL

·      The conclusion that drought stress had no significant effect is not necessarily true, due to the significant effect of the drought x salinity x sand burial interaction. This suggests there is a stronger effect of these three factors, perhaps at certain levels of drought, and needs to be further explored and discussed.

·      To my understanding, each treatment combination only had a sample size of 3. Is this really large enough to detect effects due to treatments?

ABSTRACT

·      Add a brief statement of S. alopecuorides importance to support the last sentence of the abstract

·      Line 19 – “Sand” should not be capitalized

INTRODUCTION

·      Line 36, 37 – specify that “blown-sand” is “wind-blown sand”

·      Line 41 – specify that where the soil salinity is from

·      Line 47 – is there an issue when a seed bank occurs that germinates when the sand layer thins out? Does this increase desiccation risk? Continues to state in Lines 49-52 that shallower sand burial is good. This needs to be rewritten to clarify statements.

·      Lines 54-55 – elaborate how salinity effects “ecological environment”

·      Line 56 – effects what part of photosynthetic pigments? Production? Maintenance? Please specify.

·      Lines 69-72 – where were these desert plants?

·      Line 74 – clarify what is meant by “maintain ecological balance”

·      Line 80 – what makes the desert steppes “vulnerable”?

·      Line 88-89 – “compare the intensities of these stress factors” on what?

·      Line 89 – “optimal stress” is kind of an oxymoron – is there a better way to state this if it increases germination success? Such as optimal combination of factors?

MATERIALS AND METHODS

·      Lines 233-234 – “dry all year around with little rain, dry climate”, this is a little repetitive, the annual rainfall is already given - Do you have some kind of estimate of evaporation rates? Is this from the soil?

·      Lines 235-237 – “xerophytic, medium xerophytic”, again repetitive

·      Lines 237-238 – seed weight description is a little out of place here – would perhaps make more sense if there was some description of the plant itself and its inflorescence structures (i.e., type, fruit type, etc.)

·      4.2 Experimental Design – the writing of this section is very hard to understand, lots of run-ons

·      Lines 244-249 – why were so many concentrations of sulfuric acid and water baths used? Were these treatments to explore best pre-treatment method for the seeds? This really needs clarified

·      Line 262 – what is 10 cm referring to? Was this the depth of soil for all treatments? Clarify

·      Lines 264-274 – Again, this is one big long run-on sentence that can be confusing and overwhelming.

·      Line 274-275 – Are 3 replicate plants really a large enough sample size to detect treatment effects? Generally, a sample size of at least 10 is needed.

·      Line 275-276 – were the plants all watered with fresh water or did it depend on their salinity treatment? This needs clarified

·      Figure 3 – add the treatment names (eg, D1, D2, D3…) to the figure

·      Line 280-281 – Add “trial” after “, and germination”

·      Line 284-285 – I do not understand what is meant by germination energy. This seems to be a count of the number of seeds that germinated during a peak germination period, based on the calculation. I do not see how this is a helpful variable.

·      Line 288-289 – what variables and factors were used in the ANOVA?

·      Line 290-291 – there was no description of any data collected on seedling growth. What was measured here?

RESULTS

·      Lines 100-108 – the interactions need more discussion, especially the interaction of drought. This variable alone may not be significant, but it still has a significant effect in the multi-stress combinations

·      Lines 112-122 – species name needs italicized in this paragraph

·      Line 112-113 – what kind of decreasing trend?

·      Lines 118-120 – this sentence almost sounds like data interpretation, which belongs in the discussion

·      Figure 1 – this figure is overwhelming and too busy. Perhaps break it up into three different figures?

·      Table 3 -  I am not 100% clear where this ranking came from. Please clarify

DISCUSSION

·      Line 161-162 – sentence seems out of place

·      Lines 164-168 – repetitive of introduction

·      Lines 171-172 – who found this? Is this a general statement or was it for a specific experiment?

·      Lines 173-177 – this cannot be stated indefinitely b/c energy content was not measured

·      Line 188 – “adversity” is somewhat anthropomorphizing plants. I suggest using a different term

·      Lines 196 – 207 – again, there must be some effect of drought on the other variables, leading to the significant interactions. This needs to be further explored and discussed. For example, does drought conditions exacerbate the other variables?

·      Lines 203-204 – “the negative effects of drought may be amplified”. This needs further explained based on significant interactions

·      Significant grammar issues throughout manuscript - numerous run on sentences, wrong tenses, and incorrect word choice/usage

Reviewer 2 Report

This paper discusses an interesting topic that will be of interest to researchers in the area of seed ecology.

However, there were a few minor errors in spelling, grammar and sentence structure, which are listed below:

Line 60 – Remove the word “damages” and replace with “damage”

Lines 112-122 – There are multiple occurrences of a format error for the scientific name of plant, which should be italicised.

Line 166 - Change “thick” to “deep”

Line 175 - What is meant by the term “upward digging of seedlings”?  Please express this more clearly, so it more accurately describes the way the emerging structures of the seedling move through the soil.

Line 188 - "it has been showed", please change this to ‘it has been shown’.

Line 233 - The words “dry climate” are unnecessary and can be deleted, since the description of the climate as dry is adequately communicated by the remainder of this sentence.

Line 263 - What are “full and uniform seeds”?  Does it refer to the seeds that are most likely to germinate. because the embryo is matured and the nutrient store of the seed is adequately filled? Please rephrase this section.

Lines 264-274 - The description of the methods is all one sentence and therefore a bit difficult to follow.  Please clarify this by changing the description of each treatment to single sentences.

Line 275 – What is meant by “The soil water content was watered” ?  The remainder of this sentence is also not very clear.  Perhaps it would be better to state that “Soil moisture levels of all treatments were maintained using the weighing method.”

Page 4  - Table 2 is not correctly configured.  There is a heading with no content listed below.

Also, in Materials and Methods it might be helpful to give a citation for the source of the method applied for pre-treating the seeds.  This would give some additional supporting evidence for it being an acceptable method in this study.

Other than the few issues with grammar, spelling and sentence construction identified above, this paper is quite satisfactory as an explanation of the chosen multifactorial effects on the germination of this plant.

Quality of the language and expression is good, apart from the few errors listed above.

Round 2

Reviewer 1 Report

Overall well-revised manuscript - revisions have led to more clarity of methodology and results. Just a few grammatical and clarifying suggestions. 

Line 26-28: May be more appropriate in the introductory portion of the abstract

Line 324-326: Break into two sentences

Line 341: End previous sentence at "stress" and begin a new sentence with "Therefore..."

Line 542: Add "and" before "the Petri dish..."

Line 555: End previous sentence at "(Figure 3)", delete "among them", and start "The sand burial..." as a new sentence

Line 582-587: Still confusing here - I suggest breaking it down into two sentences - one focused on the saline-alkali treatments and one on the drought treatments

Line 602: Define peak of germination
